# Dark matter searches and energy accumulation and release in materials

**Sergey Pereverzev**⋆

Lawrence Livermore National Laboratory, California, USA

⋆ pereverzev1@llnl.gov

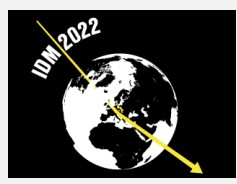

## Abstract

**Efforts to identify dark matter by detecting nuclear recoils produced by dark matter particles reveal low-energy backgrounds of unknown origin in different types of detectors. In many cases, energy accumulation and delayed burst-like releases of stored energy could provide an explanation. These dynamics follow Prigogine's ideas on systems with energy flow and the general Self-Organized Criticality scenario. We compare these models with properties of excess backgrounds in cryogenic solid-state detectors, relaxation processes in glasses and crystals, our observations of delayed luminescence in NaI(Tl), and make predictions for more phenomena present in these systems and in superconducting photon detectors and qubits. Experiments to create accurate phenomenological models are needed.**

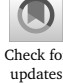

## 1 Introduction

A plausible resolution of the dark matter problem would be the direct detection of dark matter particles. Expectations are that rare nuclear recoils produced by these particles can be observed in sensitive detectors operating underground at low background radiation conditions. Low-energy recoils also can be caused by atmospheric and solar neutrinos- an effect thought to put a sensitivity limit on terrestrial searches for low-energy recoils caused by dark matter particles. On the other hand, as low-energy sensitivity improves, many detectors start to see a large number of not-well-identified low-energy events. Noble liquids dual-phase detectors can detect sub-keV recoils where very few electrons and photons are produced [1]. Recent progress in solid-state low-temperature detectors allows the detection of events with energy deposition below 10 eV, see for example background spectra for the MINER detectors in [2].

The spectra of events observed in many solid-state detectors [2, 3] and Noble Liquid detectors (some review is given in [4]) often rise sharply towards low energies, and the number of these low-energy events is larger than expected neutrino or dark matter interactions. These backgrounds rise with ionization load and are larger for detectors operating above ground. In solid-state detectors [2,3], background rises with mechanical stress in the detector material. We can assume that we are dealing with rare, condensed matter or chemical events leading to small energy releases inside our detectors. There are various ways energy can be pumped into and stored inside materials. For materials and systems out of the thermal equilibrium, where interactions of excitations, defects, and other objects or sub-systems bearing the excess energy are present, avalanche-like energy relaxation events (energy-release events) and other complex dynamic phenomena can take place. Thus, we hypothesize that stored energy relaxation events can mimic rare low-energy interactions with particles.

Systems with energy flow were first studied by Ilia Prigogine and later by self-organized criticality theory. Effects of this type remain insufficiently investigated and could be unfamiliar to the dark matter community. We provide a brief introduction and examples.

In this paper, we will discuss solid-state detectors. We present our experimental result on energy accumulation and release as delayed luminescence in NaI(Tl), make predictions of new effects in low-temperature solid-state detectors, and suggest experiments to look for relaxation avalanches at much smaller energies – down to 10 meV – in superconducting IR photon (quantum) sensors.

Possible mechanisms of energy or charge accumulation and releases in Noble Liquid dual-phase detectors are discussed in [4]. Accumulation of unextracted electrons on the liquid-gas interface can lead not only to the production of parasitic signals, but possibly to the appearance of surface charges ordering, similar to dimple crystals on liquid helium surface, and suppression of electron extraction for small signals produced below the liquid surface. This discussion will be published elsewhere [5].

## 2 Excess low-energy backgrounds

The presence of low-energy ionization events with the production of very few electrons (1, 2, 3, 4, ... 8) and the number of events rising sharply to a single electron limit was reported in many noble-liquid dual-phase detectors deployed for dark matter particles searches starting with XENON 10 experiment [6], than XENON 100 [7], XENON 1T [8], and Dark Side 50 (50 kg liquid Argon target) [9] detectors (see also Fig.2 in [4]). These ionization events cannot be caused by thermal fluctuations- ionization energies of Xe and Ar are too high. Low-energy particles cannot penetrate deep inside detectors because of self-shielding by pure Xe (Ar) liquid, and low-angle Compton scattering of gammas necessary required observation of larger-angle scattering events at a rate that is not present. On the other hand, the energy required for ionization can be already present in the detector in form of meta-stable excited molecules, chemical radicals, or trapped ions [4]. If energy or trapped charge releases can take the form of small avalanches – then the resulting event spectra could be of observed type – as we discuss below.

Energy can accumulate in the materials due to residual and cosmogenic radioactivity. Not all energy is transformed into heat and luminescence immediately following ionization events-defects, trapped ions, chemical radicals, and other long-leaved excitations can be accumulating in the materials; an increase of temperature after exposure to ionizing radiation can lead to release of accumulated energy in the form of thermally-stimulated luminescence and other related phenomena [10]. David Nygren has suggested that this stored energy is responsible for low-energy background events in the NaI(Tl) scintillators [11]. Detection of several lumi-

nescence photons during the sub-μs interval is considered an event in NaI(Tl) scintillator, and pulse-shape discrimination is used to select events resembling luminescence pulses produced by particles in DAMA-LIBRA [12] and similar experiments (see, for example, [13]).

In low-temperature solid-state detectors [2,3], researchers are looking for events that are spikes of the temperature of the target crystal, detection of hot phonon bursts by superconducting transition edge sensors placed at the target crystal surface, detection of a small current pulse when an electric field is applied to the target crystal, detection luminescence bursts with photon detectors surrounding target crystal- and combinations of these effects as additional criteria to select events produced by low-energy interactions with particles. Importantly, for many detectors, starting at energies below 1 keV, the obtained energy spectra rise sharply towards the low-energy detection threshold [2]. There is a growing amount of evidence that at least part of these events is produced by the thermo-mechanical stress in target crystals [3].

We will discuss below that mechanical stress, exposure to ionising radiation, environmental changes as temperature, pressure, electric and magnetic fields, exposure to light, IR, microwaves, sounds and vibrations can deposit excess energy into materials at low temperatures. This leads to hypothesis that presence of excess energy in material or detector and interactions between energy-bearing states can lead to correlations in delayed energy releases and excess low-energy backgrounds.

## 3 Prigogine's systems with energy flow; self-organized criticality

For a system with energy flow and away from thermal equilibrium, Ilia Prigogine [14] postulated several general principles, like the formation of dissipative systems (this can be a pattern of convection or sequence of chemical reactions), transitions from havoc to order, and generation of complexity. Prigogine ideas are important for understanding the origin of life and the functioning of the live cell. Still, they are applicable to non-organic systems where internal interactions can lead to correlations in energy dissipation processes, self-organization, self-reproduction of structures, etc. Unfortunately, for many important detector applications, interactions inside materials are not known in sufficient detail to build ab initio theoretical models.

Self-Organized Criticality theory (SOC) [15,16] analyzed the results of computer dynamics simulation for large multi-particle systems with known interactions between particles. For the system where avalanche-like relaxation takes place (a sand pile when more material is added to the top is an example) some common features can be observed: the spectrum of relaxational events (avalanches) is decreasing with event energy polynomially (not exponentially), so catastrophic events are possible; power noise spectrum is close to 1/f (pink nose), pumping energy into the system by small quanta can result in large relaxation events, and reinforcement of relaxation on small scale can lead to suppression of large relaxation events (placing the sand pile on a vibrating plate). Several examples are known in condensed matter physics where the experimental statistic is close to model calculation predictions: crack formation in materials under stress [17] and production of quantized vortexes in superconductors when the critical current density is reached [18].

Thus, when we see the spectrum of background events rising toward low energies, or a noise power spectrum close to 1/f is present, we can suspect that a SOC-like dynamic is present. Then we need to ask if energy can be stored in our material or system, what are the sources of this energy, and if energy-bearing states can interact. In experiments, we can try to increase or decrease the energy influx into our system or try quenching energy-bearing states. Pumping energy into the system by small quanta and looking for the appearance of "up-conversion-like events" is another possibility. The spontaneous transition of qubits into an excited state or the

appearance of a hot photon in a cold superconducting resonator could be examples of actual up-conversion events when energy is pumped into the materials by small (sub-gap) quanta.

## 4 Excess energy in glasses

At temperatures below the glass transition, amorphous (disordered) materials are out of thermodynamic equilibrium, and relaxation toward a lower energy state (like the formation of small crystals in a glass) can continue for hundreds of years. Relaxation processes in the glass state (response to force or other stimuli) became long and dependent on the internal state of the material, i.e., history-dependent. Relaxation can involve changing internal stress and other parameters, like dielectric constant; both applications of force or electric field can cause long relaxation processes. When the force or electric field is applied and then removed, glass can be brought into a state with higher internal energy, i.e., energy can be pumped into the system with glass-like relaxation properties by application of electric or magnetic fields, stress, etc.

At low temperatures, many subsystems in materials demonstrate complex and history-dependent relaxation properties. As examples, we can name charges and spins localized on boundaries and interphases in SQUIDs [19], the motion of charges in dielectric substrate probed by single-electron transistors on the surface [20], magnetic moments of impurities in superconductors forming spin glass [21].

Glass properties become more complex at Ultra-Low Temperatures (below mK); organic and non-organic glasses start to demonstrate the memory effect- dielectric constant "remember" the electric field in which material was cooled to low temperatures [22].

A dominant theoretical approach to glasses is based on a tunneling two-level systems model developed in the 1970s [23,24]. The TLS model is also widely used to describe noise and decoherence in superconducting sensors and qubits [25]. The TLS model postulates the existence of objects in material that are multi-particle configurations with two closely arranged minimal energy states/configurations such that tunneling in between these states/configurations is possible. Each TLS is described with two parameters: energy differences between the two adjacent minima and tunneling coupling/probability. Two-level systems were assumed to be non-interactive [23,24], and the universality of properties of different glasses originate in the universality of TLS parameters distribution in materials. The TLS models successfully describe many properties of glasses, but, as Anttony Leggett points out [26], the TLS model may not be the only model suited to describe these properties. The microscopic models for TLS with required parameters are still debated (see discussion in [25]).

In Prigogine's approach interactions between energy-bearing states lead to new emerging phenomena; interactions are the necessary conditions (but not sufficient) for avalanches in SOC model. Authors of [22] also highlighted the "importance of interactions between active defects in glasses". The above discussion demonstrates that the description of glasses and decoherence with TLS may be incomplete- it likely is missing phenomena emerging from interactions between energy-bearing states. Observation of energy up-conversion effects when energy is pumped into the material by small quanta would be important demonstration of the role of internal interactions.

## 5 Delayed luminescence in NaI(Tl)

The Saint-Gobain company, the main manufacturer of NaI(Tl) crystal scintillators, has warned on web site and in other documentation that mild exposure of crystal to UV light may lead to the appearance of low rate (few per second) scintillation pulses resembling irradiation with

few keV energy electrons, and these pulses disappear by itself in several hours or days. This looks like an up-conversion effect we expect to see for the SOC-like dynamics. As no other information is provided by the company, we tried to reproduce these experiments [27]. We also expected [4] that after-luminescence can be suppressed by exposure to red or IR light, as such exposure suppresses thermally stimulated luminescence in alkali-metal halides [28]. We have found [27] that exposure to UV light results in delayed luminescence response lasting several days, and that exposure to energetic electrons from a $^{60}$Co source leads to delayed luminescence lasting several hours, see Fig.1. We also confirmed that single energetic particle events and muons are causing a slight increase in background photon emission; to see this effect one needs to do averaging of response (traces) for many energetic particles. The delayed luminescence signals we observed after UV exposure was mostly random photon emission processes. Saint-Gobain researchers were not discussing the existence of this random flux of uncorrelated single photons and how particle-like pulsed were selected. Right after exposure to UV light delayed luminescence in our experiments was too intense to be separated onto well-defined events (photon bursts). As intensity became close to equilibrium background photon emission, more data is required to check that photon bursts present in excess of what one can expect from a random coincidence in the number of photons detected during a short time interval. We used coincidences of photon detection by right and left PMTs to trigger data acquisition and applied a pulse-shape discrimination analysis similar to the analysis used by the DAMA-LIBRA experiments [12]. The leakage into the "particle-like" domain was dependent on the left-right coincidence interval, the length of the photon counting window, and a choice of "separate event" criteria, i.e., the absence of photons for some time before and after the event. Though these cuts strongly reduce the number of "candidate particle-like events", we can see leakage into the "particle domain" [27]. A more detailed analysis with an exact replication of DAMA-LIBRA algorithms is required to quantitatively characterize this leakage under different conditions.

We have demonstrated (see Fig.1) that exposure to red light can strongly suppress delayed luminescence signals. Delayed luminescence intensity and decay time are dependent on the type of energetic particles (see also [29]); paper [30] demonstrates that the decay time of delayed luminescence after exposure to $^{60}$Co strongly depends on temperature. This suggests that other environmental factors, including pressure, electric and magnetic fields, mechanical stress, changes in the microwave or RF backgrounds, or vibrations level also can affect delayed and, possibly, fast luminescence responses.

Unfortunately, DAMA-LIBRA and other experiments trying to check /replicate observed yearly modulation in the rate of particle-like low-energy events are not collecting information about the intensity of random background photon emission. Depth of modulation (if present) and the relative phases to modulations of muon flux, solar neutrino flux, and the DAMA-LIBRA background oscillations would be of interest. In-phase variation of random photon emission with the DAMA-LIBRA oscillations could indicate the presence of a single environmental factor affecting both signals. Low-energy Solar neutrino and/or low-mass dark matter particles can have insufficient energy to produce luminescence photons. Still, it can produce hot phonons with energies sufficient to trigger the release of stored energy. A.K Drukier even suggested detection on low-energy nuclear recoils based on the release of energy stored in micro-explosives [31]. Thus, analysis of all possibilities for modulations can be complex. Our current model for the immediate and delayed luminescence responses could be insufficiently accurate: it could be missing photon bunching and/or modulations by environmental effects (could be slow seasonal temperature variations, like [30], or fast variations of atmospheric pressure during rainy seasons).

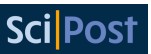

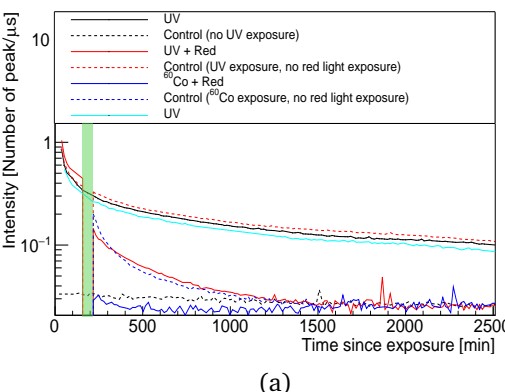

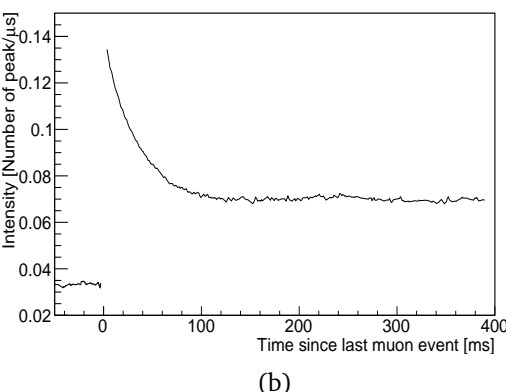

(a)
(b)

Figure 1: The delayed light intensity in NaI(Tl) [27] was monitored over time following UV exposure (a, solid black line) and Co-60 irradiation (a, dashed blue line). Three hours after the UV exposure, the crystal was exposed to red light (a, solid red line); and a pronounced decrease in the afterglow intensity was observed. The shaded green box corresponds to the red light irradiation time. Control runs were performed, in which exposure with the UV or red light sources was omitted, but the same procedure was used for source placement as in the runs with exposure. This check confirmed that the experimental procedure had a negligible contribution to the observed trend (a, dashed lines). We also observed delayed light following large energy depositions in the crystal (b). Note that we omit the data point at the zero time, which is the time of the large ionization event (b).

# 6 Phenomenology comparison

Low-temperature solid-state detectors use different single-crystal sensors like Si, Ge, SiO$_2$, CaWO$_4$, Zn, etc., cooled to 10-50 mK temperatures.

Microscopic mechanisms of small inelastic deformation (flow) of single-crystal samples were extensively studied (especially Si and Ge) at room temperatures under different load conditions, including inhomogeneous loading and micro-indentations, see, for example, [32,33]. The flow demonstrates small steps consisting of transformations in microscopic volumes of material. These transformations are changes in crystallographic structure, chemical transformations, the appearance of the twin boundaries, sliding plains, appearance and the motion of dislocations. These transformations are dissipating events and should result in heat and hot phonon productions. It is also known that the appearance and motion of defects, dislocations, etc., in dielectrics, semiconductors, and metal samples can be accompanied by emission of photons and electrons (surface electrons) from the sample [34,35]. These relaxation processes can continue after the force that causes deformation is removed. This has a strong similarity with relaxation processes in glasses. Importantly, the long relaxation process in glasses can be caused not only by inelastic deformation (flow) but by other, more "gentle" actions. An example is level-burning experiments in organic glasses at low temperatures [36]; here laser pulse is producing a notch transparency window in a glass, and longtime observation (year) of increasing absorption in this notch transparency window at low temperatures was required to demonstrate not exponential but polynomial relaxation time dependence. The non-interacting TLS model predicts that relaxation is exponential.

We know that energy accumulates in solids during exposure to ionizing radiation- and we have effects of thermally stimulated luminescence, electron emission from the surface, and thermally stimulated conductivity in irradiated samples [10]. Releases of stored energy due to

thermal activation at ambient temperature and due to tunneling should also lead to delayed luminescence, surface electron emission, and production of current carriers, or quasiparticles. We see this as delayed luminescence in NaI(Tl). Relative numbers of different energy-bearing states produced by inelastic deformation, (thermomechanical stress), or by specific types of ionizing radiation will differ. On the other hand, there are energy transfer processes between different energy-storing states and configurations, states bearing excessive energy and states responsible for photon and electron emission, and quasiparticle production. We observed that exposure to a red light suppresses both TSL and delayed luminescence, which indicates that states responsible for the energy storage in both cases are identical or have many similarities.

The intensity of TSL and TSEE in Xe crystal can decrease in order of magnitude when the crystal is annealed before irradiation [37]. One can expect similar effects of defects on TSL, TSEE, and on delayed luminescence, electron, and quasiparticle emission in other materials.

Authors of [3] introduce the term "micro fracturing" to explain the increase in particle-like events and quasiparticle production with increased thermomechanical stress in low-temperature solid-state detectors. We can state this differently: relaxation of mechanical stress includes step-like small transformation at room temperature and at low (10-50 mK) temperature.

The next questions are if relaxation processes can still contain step-like transitions in two limiting cases. One case is when the stored energy is decreasing, and the system is nearing an equilibrium state. To some extent, this is equivalent to a small number of defects produced by ionizing radiation or by UV light. The other case is lower temperatures and the presence of low-energy "excitations" - as mentioned above, many systems in materials demonstrate glass-like properties at low temperatures. Our tentative answers are "yes" in both cases. These answers follow Prigogine's ideas and SOC theory. It could be possible to check these predictions experimentally.

One possibility is to check for photon bunching in NaI(Tl) delayed luminescence after exposure to UV light when the material is closing to the equilibrium (low background luminescence) state, we have discussed this above. In low-temperature solid-state detectors, one can look for delayed signals appearing after exposure to ionizing radiation, UV or visible light. We expect that bursts of photon, phonon, and quasiparticle emission could be present. These experiments are better to try on samples with low mechanical stress after long storage at low temperatures in a low radioactivity underground laboratory. Low-temperature experiments with NaI(Tl) where both luminescence and temperature spikes are recorded are of high interest as they could demonstrate an absence of keV scale heat production when small photon emission bursts are detected.

# 7 Testing for SOC-like dynamics in superconducting devices

Superconducting nanowire single photon detector (SNSPD) is a new rapidly developing IR photon sensors technology demonstrating in comparison with other superconducting photon sensors technologies the fastest photon counting rate, the lowest dark counts, along with high energy sensitivity. Absorption of an IR photon in a superconducting nanowire carrying small DC results in breaking superconductivity in a small section of the nanowire and heat production in this normal section by the applied current. The normal section of the nanowire wire starts to grow, and a voltage peak appears across the nanowire. Importantly, there is no energy dissipation inside the superconducting nanowire by the small readout current while the detector is waiting for a photon, and thermo-mechanical stress energy is small and likely relaxation time after cooling down is fast coming to stop for thin (several nm) and narrow nanowire. This means that energy pumping into nanowire and substrate vanishes when this detector is

waiting for a photon arrival, so SOC-like dynamic should be suppressed in these devices (we discussed this idea in [38]). The low dark count rate allows us to look for SOC-like dynamics and effective energy up-conversion effects in these devices: one can expose the SNSPD to a low-intensity flux of microwave photons with photon energy below the superconducting gap in the nanowire. Then photons cannot break Cooper pairs in nanowires and can only produce low-energy excitations in a substrate or sub-gap excitations in the superconductor. An increase in the dark counts would indicate energy up-conversion-like events in the device, and observation of dark count dependence on the frequency of microwave radiation can provide some spectroscopy of low-energy states in the device materials which are involved in SOC-like dynamics. One can change electron concentration in a nanowire by applying an electric field perpendicular to the substrate, and this will affect the superconducting transition temperature and critical current. By lowering the temperature and critical current simultaneously, one can try to increase the energy sensitivity of SNSPD. This also will make tests for SOC-like dynamics more sensitive. Photon detection at longer wavelengths is interesting for space microwave astronomy and searches for axions.

## 8 Discussion

We compared properties of excess low-energy backgrounds in dark matter detectors, relaxation processes in glasses, and Prigogine's ideas about systems with energy flow. We came to the hypothesis that processes of energy accumulation by "interacting excitations" in materials and releases of this stored energy can be present in all detectors and cause events of burst-like emission of photons, phonons, or quasiparticles, emission of surface electrons, etc. At the same time, our current knowledge of these processes and internal interactions are insufficient for model-independent dark matter searches: we may not be able to exclude beyond a reasonable doubt these processes as an explanation for observed low-energy events in our detectors.

In other words, a slight shift of paradigm is required. In addition to nuclear and particle physics effects that produce the dominant part of backgrounds in dark matter particle searches, we need to pay attention also to possible non-equilibrium thermodynamic effects. Intuitive expectations based on equilibrium thermodynamics are not working well here: we are custom to energy transfer cascades from high-energy excitations to low-energy excitations, but here we can find cascades producing large energy events or excitations from small energy excitations.

We have insufficient knowledge of excitations in materials and their interactions for ab initio calculations/modeling dynamics of complex systems away from thermal equilibrium. Moreover, we expect to find new effects based on the comparison of phenomenology we see in different systems. Looking for these new effects will help to build more accurate phenomenological models and chart boundaries where simplified models can work. A correct understanding of material processes should help to select better materials and readout techniques to improve detectors' sensitivity for elastic coherent neutrino scattering and dark matter particles, or/and improve limitations that direct detection experiments put on dark matter particle models.

It is commonly accepted that muon or other energetic particle can cause correlated errors in multi-qubit superconducting quantum processors. Excess backgrounds events described in [2,3] have sufficient energy to cause quantum errors and correlated quantum errors: these events emit energetic phonons and photons, and relaxation avalanches could be non-local. Moreover, microwave pulses used to control qubits are depositing energy into materials, thus potentially leading to delayer relaxation events and possibly up-conversion events. Probabilities of such events could be studied experimentally for different materials and device designs to improve coherence time of QI devises.

Our general conclusion is that searches for low-energy interactions with neutrinos or hypothetical dark matter particles could be more efficient if we in parallel acquire and analyze data on energy accumulation and release effects in materials and devices. Studies of non-equilibrium thermodynamics effects using tools of particle physics, condensed matter, and QI techniques would benefit all these disciplines. This invites wider collaboration between HEP and condensed matter scientists, and we argue that joint research programs between funding agencies (like HEP and BES divisions of the DOE) are required.

## Acknowledgements

This project is supported by the U.S. Department of Energy (DOE) Office of Science/High Energy Physics under Work Proposal Number SCW1508 awarded to Lawrence Livermore National Laboratory (LLNL). LLNL is operated by Lawrence Livermore National Security, LLC, for the DOE, National Nuclear Security Administration (NNSA) under Contract DE-AC52-07NA27344. LLNL-PROC-840806.

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
