# Peer review of "Dark matter searches and energy accumulation and release in materials"

_SciPost Physics Proceedings, doi:SciPost Phys. Proc. 12, 009 (2023)_

## Round 1 · Referee Report · Anonymous (Referee 1) · 2022-11-24

Strengths

•I like the proposed idea that excesses in solid-state detectors could be related to metastable states that cause avalanche effects. This could potentially predict the power-law energy spectrum and certainly is an important first step in building models for the excesses observed with cryogenic solid-state detectors.
• The parallels with the theory of self-organized criticality are very interesting, especially because this theory seems to appropriately model the behavior of earthquakes (Gutenberg-Richter law), and a parallel to relaxation effects in solids in thinkable.
• The argument that delayed luminescence could lead to delayed increases of the event rate in NaI(Tl) detectors, caused by an environmental impact that happened already earlier, is interesting and if this was not studied previously, it could be a nice check to perform for experiments using NaI(Tl).

Weaknesses

• The author is unspecific about which excesses he wants to explain in this work, and switches between the wording “all detectors” and the DAMA excess specifically. While some of the referenced excesses can have a common origin (cryogenic solid-state detectors), others were already shown to have qualitatively different properties (CCD detectors, gaseous targets). This was shown in the references the author cites. How the DAMA excess, which is mainly characterized by its annual modulation, is related to the others is not clear to me and not motivated in the manuscript in way that I could comprehend.
• The proposed theory is a purely heuristic model and does not include any concrete physics (which states and excitations, etc). Thus, it does not constrain any existing hypothesis for the origin of the excesses, nor does it build a predictive and testable model.
• There are some major issues with scientific reasoning. The author generalizes from some sources to others without argument, draws conclusions without basing them on experimental evidence or calculations, and does not reference some of his statements (see more detailed comments below).
• The author explains the time-dependency in the event rate of low energy excesses with the physics of glasses, and the power law-ish energy spectrum with the theory of self-organized criticality. The connection between these two phenomena is not clear to me and the author does not provide any evidence that either of the phenomena is realized in the nature of the detectors observing the excesses.
• While the studies of delayed luminescence are certainly interesting, the connection to the modulation in the DAMA event rate is not properly argued and attributed to “environmental effects”.
• The structure of the text is more that of a speech, than that of a scientific paper, which makes it very hard to read.

Report

The author discusses low energy excess backgrounds observed by multiple low threshold experiments. He relates the phenomena to the theory of self-organized criticality and the physics of glasses. He also discusses the modulation in the event rate observed by the DAMA experiment and relates it to the phenomenen of delayed luminescence.

In addition to the listed strengths and weaknesses, I have some detailed comments. Due to the lack of line numbering I can only reference parts of the manuscript and not individual sentences, I hope the comments are still useful.
• Liquid noble detectors are mentioned in the introduction but then never again.
• Reference [5]: As I understand “Scientific American” is a popular science journal. Please reference scientific information from scientific journals.
• Headline of section 2: please use shorter headlines
• In several places in the manuscript are spaces missing, this should be easy to fix with a spellchecking program.
• Beginning of section 2: I think 10-1 eV should mean 10-1000 eV.
• Right after this sentence: The claim, that many experiments observe an increase of background with thermomechanical stress dearly needs references. To the best of my knowledge, this is at the moment a hypothesis and only confirmed in one measurement (Ref. [7] in the manuscript). In this paragraph, the author needs to be very specific about which observations were made in which measurements, instead of generalizing from one measurement to an unspecified number of excesses.
• “Thus, the low-energy background we see is a natural relaxation process of mechanical stress release in single crystals and solids in general.” This is a hypothesis but presented in the manuscript as if it was a result.
• Beginning of section 3: are the used target materials all amorphous? To the best of my knowledge the target crystals are monocristalline. Please specify for which detectors the theory of glasses is applicable. Also, please add some references for the statements in this paragraph.
• Ref. [14, 15, 16]: They are listed together with phenomena that seem unrelated to the content of the text above and below. Please specify how they are relevant for the arguments.
• Section 3, start of third paragraph: I don’t see on which arguments the conclusion that is drawn here is based. Please explain in detail and avoid basing scientific statements on “parallels and analogies”.
• Section 4: This section seems very important in the line of arguments that interpret the experimental results that are presented later on. However, there is not a single reference listed in this section, and the statements that are made are not trivial.
• “The author posits that exposure of the solid-state detector to ionizing radiation or UV light should lead to delayed low-energy background events with events spectrum and decay time resembling those caused by mechanical stress.” This hypothesis was rejected for at least one experiment (arXiv:2207.09375), that does not see an increase in the low energy background after intervals of strong exposure to calibration sources. Please specify which excesses should be explained by this.
• Section 5: This section contains very interesting information about current open questions about glasses. However, the relevance of this information to the rest of the manuscript is not clear to me and should be explained in more detail.
• Section 6: Please describe your experimental setup, your analysis chain, and your results in more detail, quantitatively and in this order. Were the experiment and analysis performed by yourself alone? Otherwise, I recommend referencing/acknowledging collaborators.
• Section 6: A reference to the criticized Saint-Gobain analysis would be helpful.
• Section 6: “The photon flux detected by DAMA-LIBRA and similar experiments consists of delayed luminescence photons (mostly random) and bursts of photon emission produced as a fast/immediate response to external particles.” Is this a result or a hypothesis? If the former, then please reference, if the latter, please formulate as such.
• Section 6: “This suggests that other environmental factors also can affect delayed luminescence response and cause modulation of the DAMA-LIBRA signal; the time of the maximum intensity of random/uncorrelated delayed luminescence can be different from the maximum of muon flux which pumps energy into the system (this was not yet checked).” This argument seems to be one of the most central ones of the manuscript and deserves some more discussion. Which environmental impacts? Could the delay really shift the peak of the event rate? How would this work? In your experimental results the delayed luminescence is released on an exponential scale after the UV irradiation, how would that shift a peak?
• Section 7: The conclusion does mostly state that further work needs to be done, but does not summarize the results, statements, and significance of the work at hand. I strongly recommend adding this. In the last sentence, the decoherence in quantum sensors is mentioned for the first time in the manuscript, I would either explain this claim earlier or remove it.
• Page 6: I recommend putting the figures in the position in the document where they are discussed, not at the end of the document. Also, a smaller figure size would still transport all the information but free up space for text.
• Fig 1: Was the figure made specifically for this paper or published elsewhere? In the latter case, please reference.
• Fig 1: I don’t understand the second plot (b). What decay time is expected of the scintillation light pulses from this target? Are these pulses from particle hits not always several tens of milliseconds long? How can a statistic be made for the number of pulses per mus? Are these pulses really so fast?
• References: I recommend using BibTex or a similar references manager. Otherwise, please implement common formatting, indentations, border width, etc.

I summarize that the intent of this work is certainly important and the ideas have the potential to make a difference in the field. The current state of the manuscript is hard to comprehend, contains incorrectly generalized and unreferenced statements that could cause dangerous confusion among readers, and assertions that are not fully based on the scientific method. I recommend a comprehensive revision based on the below suggestions and a subsequent second round of review.

Requested changes

• I recommend adopting a structure that is easier to comprehend for readers. I especially suggest:
⁃ Start the introduction by specifically describing the problem (which excesses, their properties, references), explain that no expected phenomenon fits the observations (can’t be radiation backgrounds, dark matter, …), motivate your explanation for them, then give an overview of the sections in the paper and what you will discuss/show there.
⁃ After the introduction, first explain the theory (self-organized criticality, properties of glasses), and which behavior of the detectors/excesses it would predict. Do this in a scientific way: make predictions that are testable with data.
⁃ Then continue with your sections, discussing individual observations of experiments and their fit to your theory in detail.
⁃ End with a conclusion where you summarize the results, statements, and impact of this work.
• Please reference the individual experiments that observe the excesses.
• Statements/results need to be clearly distinguishable from assumptions/hypotheses. Former ones have to be referenced properly.
• The detailed comments from above should be addressed.
• The content of the manuscript is very advanced and brings together concepts from different fields. The more important is the preparation of this information in a way that is accessible to readers without extensive previous knowledge. Adding more figures that describe the discussed phenomena could make the reading easier.
• Please add line numbers for the next round of review.

---

## Round 2 · Referee Report · Anonymous · 2023-1-25

Strengths

The author formulates interesting ideas to build models for excesses in dark matter and other experiments.

Weaknesses

For future work, I suggest the author should put their ideas either into a quantitative model that can be tested with data, or perform experiments that test the hypothesized effects qualitatively.

Report

The author has implemented all comments sufficiently and made a clearly visible effort to improve the clarity of the paper. The second version of the manuscript is in a state that can in my opinion be published as proceedings.

---

## Round 2 · List of Changes

First, I need to thank the referee for his/her thoughtful and helpful comments and suggestions.

Required changes:
My main goal is to make this paper clear and easy to understand- and this is a difficult task for a limited paper volume and method I need to use to come to conclusions. There is a common problem I can see in all low-energy threshold detectors looking for dark matter particles: they all have low-energy background events with the number of events rising to lower energy (a form of the spectrum), and they are not nuclear recoils caused by atmospheric or solar neutrinos or by dark matter particles, because there are too many of these events, and a number of these events is rising with energy which is pumped into detector’s material. My hypothesis is that in different detectors we see a process of energy pumping in materials/detectors and delayed release events, while release events can be avalanche-like and mimic interactions with particles [4].
In different detectors, we see different features of this type of process, and we have no solid experimental proof of this dynamics for a single detector; only a large picture points out that we have a process of this type.
So, I need to refer to a large number of papers – about Noble Liquid detectors, NaI(Tl) detectors, and Solid State low-temperature detectors. I am referring separately to different noble liquid detector projects and DAMA -LIBRA projects. For solid-state detectors several recent Excess workshops collect data in a single place- so I refer to workshop publications and one “review” paper describing results from many projects -references [2,3]. I am afraid, that currently there are no separate publications from each of these projects which describe properties of excess low-energy background; so, I need to refer readers to these two papers.
Physical mechanisms of energy/charge storage and interactions between excitations/energy/charge bearing states are very specific for Noble Liquid detectors, so I refer readers to my paper [4] and to publication in the proceeding of much more special Noble Elements conference [5]. (I have not submitted this paper to arXiv yet but will do.)
I tried to rewrite the paper following the referee’s suggestion. I also tried to point out clearly where I am hypothesizing, and I suggested experiments to check these hypotheses. It would be good to add/reproduce several pictures from other publications to make some points easier for readers to understand, but the size of my paper is already large. Would the editor allow it, I will be glad to add a couple of pictures.
I added line numbers.

Detailed comments:
• Liquid noble detectors are mentioned in the introduction but then never again.
I explained above- it need separate paper(s)

• Reference [5]: As I understand “Scientific American” is a popular science journal. Please reference scientific information from scientific journals
Sorry,” Scientific American” often publishes high-quality review papers.

• Headline of section 2: please use shorter headlines
I changed headlines

• In several places in the manuscript are spaces missing, this should be easy to fix with a spellchecking program.
• Beginning of section 2: I think 10-1 eV should mean 10-1000 eV.
Solid state detectors indeed detect 1-10eV energy depositions

• Right after this sentence: The claim, that many experiments observe an increase of background with thermomechanical stress dearly needs references. To the best of my knowledge, this is at the moment a hypothesis and only confirmed in one measurement (Ref. [7] in the manuscript). In this paragraph, the author needs to be very specific about which observations were made in which measurements, instead of generalizing from one measurement to an unspecified number of excesses.
Ref. [3]( in rewrite paper) is bringing examples, new experiments

• “Thus, the low-energy background we see is a natural relaxation process of mechanical stress release in single crystals and solids in general.” This is a hypothesis but presented in the manuscript as if it was a result.
I fixed this. The actual hypothesis is that at room temperature and at low temperature relaxation of mechanical stress leads to photon, phonon and quasiparticle emission and can lead to bursst of photon. phonon, and quasiparticles emission
• Beginning of section 3: are the used target materials all amorphous? To the best of my knowledge the target crystals are monocristalline. Please specify for which detectors the theory of glasses is applicable. Also, please add some references for the statements in this paragraph.
In new version, I separated discussions of glasses (disordered solids) and solid-state low-temperature detectors which use single-crystal targets.
I added more glass properties and effects, though not everywhere I can find good references.

• Ref. [14, 15, 16]: They are listed together with phenomena that seem unrelated to the content of the text above and below. Please specify how they are relevant for the arguments.
The general statement is that at low temperatures we generally see more glass-like relaxation/behavior in all materials, including single crystals. This is due to “subsystems” like defects, charges, magnetic moments,etc. I illustrated this statement by examples; these examples are, in fact, are important for understanding different noise mechanisms

• Section 3, start of third paragraph: I don’t see on which arguments the conclusion that is drawn here is based. Please explain in detail and avoid basing scientific statements on “parallels and analogies”.
When you apply force or an electric field to a material with glass-like relaxation properties, it starts to relax toward a minimal energy state for the case of applied force or field. When you remove the force or field, the material is further away from the minimal energy state for zero force or field, i.e., you deposit some extra energy into the material. It can take a long time for this energy to “go out”.

• Section 4: This section seems very important in the line of arguments that interpret the experimental results that are presented later on. However, there is not a single reference listed in this section, and the statements that are made are not trivial.
In a new version, I provide a reference on a book [10] that describes temperature-stimulated luminescence and electron emission in materials exposed to ionizing radiation. These effects demonstrate that energy is accumulating in materials under irradiation.

• “The author posits that exposure of the solid-state detector to ionizing radiation or UV light should lead to delayed low-energy background events with events spectrum and decay time resembling those caused by mechanical stress.” This hypothesis was rejected for at least one experiment (arXiv:2207.09375), that does not see an increase in the low energy background after intervals of strong exposure to calibration sources. Please specify which excesses should be explained by this.
Paper (arXiv:2207.09375) describes a rather strong/high-energy events background. “Mechanical energy” (defects) can be pumped into the crystal during cool-down due to “suspension” and due to thermal gradients inside a large crystal. Was heat exchange gas used during cool-down? Another suspect is the different contraction of the crystal and TES sensor; in this case, excess energy is delivered directly into the sensor. When the background is already large, it could be difficult to see a small change.

• Section 5: This section contains very interesting information about current open questions about glasses. However, the relevance of this information to the rest of the manuscript is not clear to me and should be explained in more detail.
TLS model is not expecting energy-up conversion events, which could be present. So, I am suggesting to look for these events.

• Section 6: Please describe your experimental setup, your analysis chain, and your results in more detail, quantitatively and in this order. Were the experiment and analysis performed by yourself alone? Otherwise, I recommend referencing/acknowledging collaborators.
Reference [28] is provided.

• Section 6: A reference to the criticized Saint-Gobain analysis would be helpful.-added

• Section 6: “The photon flux detected by DAMA-LIBRA and similar experiments consists of delayed luminescence photons (mostly random) and bursts of photon emission produced as a fast/immediate response to external particles.” Is this a result or a hypothesis? If the former, then please reference, if the latter, please formulate as such.
When we see a flux of uncorrelated photons coming out of NaI(Tl), there should be some energy source for this. This could be residual stress, but this energy should disappear over time. Energy influx due to muons and residual radioactivity is always present. TSL is known for NaI(Tl), so we know that radioactivity is pumping energy into the material. When we add more pumping, we see an increase in random photon flux. As for pulses- they could be real, i.e., produced by particles, and fake- produced by correlations in energy releases due to interactions of states storing energy. So, the hypothesis here is that “fake pulses” can be present.

• Section 6: “This suggests that other environmental factors also can affect delayed luminescence response and cause modulation of the DAMA-LIBRA signal; the time of the maximum intensity of random/uncorrelated delayed luminescence can be different from the maximum of muon flux which pumps energy into the system (this was not yet checked).” This argument seems to be one of the most central ones of the manuscript and deserves some more discussion. Which environmental impacts? Could the delay really shift the peak of the event rate? How would this work? In your experimental results the delayed luminescence is released on an exponential scale after the UV irradiation, how would that shift a peak?
Particles produce an immediate response and delayed luminescence. Both components can depend on temperature, pressure, electric and magnetic fields, presence of IR or microwave radiation. Photon bunching in delayed luminescence also can depend on these factors. Muon flux is pumping energy into material and is modulated. Solar neutrinos can produce hot phonons and trigger stoa red energy release; solar neutrino flux is modulated. The main point is that we do not know the material response in sufficient detail.

• Section 7: The conclusion does mostly state that further work needs to be done, but does not summarize the results, statements, and significance of the work at hand. I strongly recommend adding this. In the last sentence, the decoherence in quantum sensors is mentioned for the first time in the manuscript, I would either explain this claim earlier or remove it.
Energetic particles can cause correlated errors in quantum processors, but spontaneous energy release even of 1 eV scale is also sufficient to cause a qubit transition to the excited state. We use microwave signals to control qubits. Microwave photon energy is below the gap in a superconductor. But microwave photons can pump energy into materials, so energy up-conversion events would be a problem.

• Page 6: I recommend putting the figures in the position in the document where they are discussed, not at the end of the document. Also, a smaller figure size would still transport all the information but free up space for text.
-done

• Fig 1: Was the figure made specifically for this paper or published elsewhere? In the latter case, please reference
-provided.

• Fig 1: I don’t understand the second plot (b). What decay time is expected of the scintillation light pulses from this target? Are these pulses from particle hits not always several tens of milliseconds long? How can a statistic be made for the number of pulses per mus? Are these pulses really so fast?
-explained

• References: I recommend using BibTex or a similar references manager. Otherwise, please implement common formatting, indentations, border width, etc.
I improve the formatting (sorry, still working to learn using the reference manager…)

I tried to rewrite the paper following the referee’s recommendations.
Please, provide me feedback if it is easier to read now.
Thanks,
Sergey Pereverzev

You are currently on this page

Resubmission scipost_202210_00066v2 on 21 December 2022

---

## Editorial Decision

published